# Embedded-AMP: A Multi-Thread Computational Method for the Systematic Identification of Antimicrobial Peptides Embedded in Proteome Sequences

**DOI:** 10.3390/antibiotics12010139

**Published:** 2023-01-10

**Authors:** Germán Meléndrez Carballo, Karen Guerrero Vázquez, Luis A. García-González, Gabriel Del Rio, Carlos A. Brizuela

**Affiliations:** 1Computer Science Department, CICESE Research Center, Ensenada 22860, Mexico; 2School of Mathematical & Statistical Sciences, University of Galway, H91 TK33 Galway, Ireland; 3Department of Biochemistry and Structural Biology, Instituto de Fisiologia Celular, UNAM, Mexico City 04510, Mexico

**Keywords:** antimicrobial peptides, embedded peptides, machine learning, autophagy, longevity

## Abstract

Antimicrobial peptides (AMPs) have gained the attention of the research community for being an alternative to conventional antimicrobials to fight antibiotic resistance and for displaying other pharmacologically relevant activities, such as cell penetration, autophagy induction, immunomodulation, among others. The identification of AMPs had been accomplished by combining computational and experimental approaches and have been mostly restricted to self-contained peptides despite accumulated evidence indicating AMPs may be found embedded within proteins, the functions of which are not necessarily associated with antimicrobials. To address this limitation, we propose a machine-learning (ML)-based pipeline to identify AMPs that are embedded in proteomes. Our method performs an in-silico digestion of every protein in the proteome to generate unique *k*-mers of different lengths, computes a set of molecular descriptors for each *k*-mer, and performs an antimicrobial activity prediction. To show the efficiency of the method we used the shrimp proteome, and the pipeline analyzed all *k*-mers between 10 and 60 amino acids in length to predict all AMPs in less than 20 min. As an application example we predicted AMPs in different rodents (common cuy, common rat, and naked mole rat) with different reported longevities and found a relation between species longevity and the number of predicted AMPs. The analysis shows as the longevity of the species is higher, the number of predicted AMPs is also higher. The pipeline is available as a web service.

## 1. Introduction

Antimicrobial peptides (AMP) represent an important tool to study the natural defense used by living organisms. These peptides are used by all living forms to control the spread of bacteria [1], fungi [2], viruses [3], and other parasites [4]. While AMPs kill microbial cells, at low concentration these peptides exert other activities that are important for the well-being of living forms. For instance, AMPs induce autophagy [5], a catabolic cellular process relevant for cellular homeostasis and health [6]. The ability of AMPs to induce autophagy has led to the development of new pharmaceuticals able to control the spread of multi-drug resistant bacteria, such as *Mycobacterium tuberculosis* [7,8].

AMPs are generally encoded and produced as single peptides; however, there are several examples of antimicrobial peptides that are embedded within pro-peptides or proteins. For instance, the primordial role of *Saccharomyces cerevisiae* in alcoholic fermentation is the result of these cells secreting a group of AMPs to kill other competing fungi; those AMPs are derived from the proteolysis of the glyceraldehyde 3-phosphate dehydrogenase protein [9]. Furthermore, autophagy may lead to the proteolysis of human cytosolic proteins to render AMPs into the lumen of autophagosome, endowing this organelle with antimicrobial properties [10].

Due to the relevance of AMPs in health and for their contribution in understanding the mechanisms of life, diverse databases have been developed to organize all the available information about them [11]. This accumulation of data has led to the development of computer-based methods aimed to assist researchers in the identification or design of novel AMPs [12,13,14]. Any of these methods are trained with known examples of AMPs, and these methods commonly use sequence-derived peptide features, such as sequence composition and physicochemical properties, among others; to date, the most effective methods to predict AMPs are those based on shallow machine-learning algorithms [15,16].

To computationally identify AMPs found in living organisms, the sequence data usually comes from genomes, transcriptomes, and/or derived proteomes. Few methods have been reported to use such input data to identify AMPs. For instance, ampir is an R package that predicts AMPs from genomes [17]; Macrel is a pipeline to predict AMP from metagenomes [18]; different methods have been applied to identify AMPs from transcriptomes [19,20,21,22]. In all these cases, the methods consider only genes or transcripts that code for peptides; no method has been reported to identify AMPs embedded into proteins.

In the present work, we describe a computer-based method aimed to identify embedded AMP into proteins. To that end, we implemented and evaluated a multi-threaded code to identify non-redundant peptides of different lengths (*k*-mers). The identification of AMPs is performed by a machine-learning (ML) model trained on a dataset of known AMPs. As an illustrative example on how this new tool may assist researchers to generate a hypothesis, we apply the developed pipeline to study the relationship between the number of predicted AMPs and longevity. Since the induction of autophagy has been reported to extend the lifespans of different species [23], and as noted before, AMPs are capable of inducing autophagy or are produced during autophagy induction, we hypothesize long-lived species are more likely to embed AMPs in their proteomes. By applying this tool to the analysis of mammals with previously reported longevity and proteome, we observed rodents with longer longevity (naked mole rats) tend to present higher proportions of AMPs embedded in their proteomes than shorter-lived rodents (common rat and common cuy). Thus, we describe a computational method that allows for the identification of potential AMPs embedded within proteins and provide examples where such a methodology may be useful.

## 2. Results

### 2.1. Computation Efficiency

The average computation time of four different implementations of the proposed algorithm for the task of processing the shrimp proteome is presented in Figure 1. The basic workflow of the proposed algorithm implies five steps: read the proteome (18,694 sequences), remove duplicates (2657 sequences), identify unique *k*-mers (10 to 60-mers), compute 51 molecular descriptors, and apply a pre-trained support vector machine model. The four different implementations are the result of applying two different libraries for building the SVM models (in-house implementation of SVM and LibSVM [24] and two different compilers (Intel compiler [25] and G++ compiler [26]). The experiment was executed five times for each implementation in 1, 2, 4, 8, 16, and 24 (maximum number of threads) threads. 

In general, it can be observed the in-house implementation of the SVM outperforms the original LibSVM versions, which is the Intel compiler, the implementation with the fastest execution time for each number of cores. Each time the number of cores is duplicated, the execution time of each implementation is reduced by half or close to it; this is an indicator of an efficient use of the computational resources. 

More precisely, we conclude the in-house SVM implementations are faster than LibSVMs independently of the compiler and the number of cores used. The differences in the execution times between each implementation diminished as the number of threads increases and practically disappeared with 16 and 24 threads. However, the implementation with the lowest execution time (in-house SVM, Intel C++ compiler) is at least two times faster than the rest of the implementations for the six scenarios tested (number of threads used).

### 2.2. Scalability and Computational Efficiency 

The four implementations developed in this work are analyzed in Figure 2. Figure 2A shows the speedup achieved by the four implementations. Speedup is defined as $S_{t}=T_{1}/T_{t}$ where $T_{1}$ is the execution time of the implementation for one thread, while $T_{t}$ is the execution time of the implementation for *t* threads. Figure 2A shows a desirable speedup, which is a linear speedup. The proposed algorithm is scalable; it can be observed as the number of threads increases, the four implementations scale well up to 24 threads, which are the LibSVM versions of implementations with better speedup. This can be caused by the higher execution time of this implementation for a single thread. Figure 2B shows the computational efficiency for the four implementations. Efficiency measures the proportion of time a processor is effectively used and is defined as $E_{t}=S_{t}/t$, where $S_{t}$ is the speedup achieved for *t* threads. In general, the four implementations are highly efficient; they achieved efficiencies greater than or equal to 87%. LibSVM implementations achieved the higher efficiency, close to 92%, while the in-house SVM intel compiler implementation achieved the lowest efficiency, 87%. As mentioned earlier, this is caused by the higher execution time taken for LibSVM implementation on a single thread.

### 2.3. Peptide Sequence Frequency 

The size distribution of the protein sequences in the putative proteome from shrimp is presented in Figure 3A. The total number of different proteins included in this proteome was 18,694, and the total length for all protein sequences included 12,580,817 amino acid residues, and 55% of the proteins have lengths below 500 amino acids.

### 2.4. Testing Predictions of Embedded AMP on Shrimp Proteome

The predictions of AMPs from the shrimp’s proteome are presented in Figure 4. We observe the larger the *k*-mer, the lower the probability to be predicted as AMPs or non-AMPs, while most predictions for AMPs and non-AMPs fall in the range of 20–30 amino acids in length. Interestingly, the AMP dataset used accumulates most AMPs in peptides of around 20 amino acids in length, while for non-AMPS the distribution accumulates peptides in length below 60 residues.

### 2.5. Predicting AMP in Different Animals with Different Longevities

Considering AMPs have been associated with autophagy induction [27] and autophagy induction has been shown to increment lifespan in many different species [28], we decided to analyze the proteomes of mammals with reported longevity at the AnAge database [29]. The longevity and mass of each animal included in this database is presented in Figure 5; the graph also displays the best linear adjustment for this data where longevity (Tmax) is a function of the natural logarithm (Ln) of animal mass in grams (M):Tmax = −0.4742 + 2.176∗Ln(M)(1)

The heavier the animal, the longer it is expected to live.

We kept only those animals whose proteome sequences have been reported (see Table 1). We noticed Rodentia presented instances in animals with high, medium, and low longevity; thus it represents a good model to compare the frequency of AMPs predicted in those animals. Among the Rodentia order, we noticed *Heterocephalus glaber*, the naked mole rat, falls out of the linear trend of expected longevity and is among the long-lived animals (see red circle in Figure 5). For comparison, *H. glaber* lives up to thirty years, while *Rattus norvegicus* only lives up to four years; yet, *R. norvegicus* adults reach up to 300 g in weight and *H. glaber* adults up to 35 g. Interestingly, more AMPs were found embedded in the proteome of *H. glaber* (22,221) than in the *R. norvegicus* (14,087), despite the latter having more proteins (34,707) than the naked mole rat (32,702). To verify these findings were not the consequence of the naked mole rat having longer proteins than the rat, we computed the total number of amino acid residues in each proteome and found the naked mole rat had less amino acid residues (20,603,935) than the rat (23,307,820). A similar trend is observed for *Cavia porcellus*, the common cuy, which has a medium longevity, and it presents more predicted AMPs than the common rat but less than the naked mole rat. In consequence, the AMPs/protein length ratio is not related to longevity, only the AMP frequency.

### 2.6. Evaluating Predictions of Embedded AMPs with Known Cases

We noted previous works have reported AMPs embedded in proteins. To validate the accuracy of our predictions, we compared the predictions achieved by our method with the embedded AMPs found on the three isoforms of the glyceraldehyde 3-phosphate dehydrogenase (GAPDH) protein [9]. Our results summarized in Appendix A show only 10 out of the 16 embedded AMPs were predicted as AMPs (true-positive rate = 62.5%); interestingly, the GAPDH isoforms contained stretches of sequences where multiple AMPs were predicted (indicated as concatenated in the table). A more recent work showed evidence that some peptides derived from the yeast proteome could present AMP activity; yet, no experimental evidence for AMP activity was presented for individual peptides [30]. Assuming all those peptides were AMPs, our method predicted seven out of the seventeen peptides identified as AMPs (true-positive rate = 41.17%) in that study (see Appendix A); in this case, several true AMPs were embedded within larger predicted AMPs as opposed to the concatenated AMPs observed in the GAPDH protein. Thus, the true-positive rate of our method for these two cases was 17/33 = 51.5%. In contrast, the false-positive rate of our method for these two test sets was 16/33 = 48.48%.

## 3. Discussion

We report a concurrent implementation of a method useful at identifying antimicrobial peptides (AMP) that are embedded within proteins at a proteome scale. Our results show the time required to identify unique *k*-mers of different lengths (10 to 60 amino acids in length) in our computer cluster takes about 20 min using 24 threads. The computation of the AMP predictions on the other hand, since we used a web service, varied depending on different factors that are out of our control; hence, we do not report any time efficiency for that computation. The web service can be accessed at https://biocom-ampdiscover.cicese.mx (accessed on 4 January 2023). On this site, the user can upload its proteome, and the system will provide one file for each *k* containing the *k*-mers predicted as AMPs; also, a file containing the pruned proteome will be an output of the predictor. Notice the main contribution of our work is not the ML approach but an efficient and scalable algorithm for the in-silico digestion of proteomes.

We provide two examples on the use of our method to predict AMPs embedded into proteomes. The first example aimed to show the distribution of embedded AMPs in shrimp proteome. We noticed the likelihood to be predicted as AMPs is related to the *k*-mer length, with the *k*-mers of length like those used to train the SVM algorithm (20 amino acids in length) having the largest likelihood. It is common to find training dataset properties are learned by machine-learning methods, inducing an unintentional bias in the predictions [31,32]. Considering that bias, the proportion of unique 20-mer AMPs embedded in shrimp’s proteins is 8.5% (43,487,577 predicted AMP out of 513,477,690 total *k*-mers). To reduce the number of predicted AMPs, we used a web service that combines four machine-learning algorithms trained to predict AMPs. That service reduced by 80% the total number of AMPs predicted (see Figure 3B), rendering 8,697,515 embedded unique 20-mer AMPs in the shrimp proteome. Our code does not consider if these peptides are generated by the available proteases in a species; hence, our predictions are an over-estimation of the AMP arsenal that an organism may express at any given time. Yet, the predictions may be useful for (i) estimating the likelihood of finding embedded AMPs within a proteome and (ii) discovering novel AMP sequences that may emerge upon the presence of novel protease activities. For instance, it has been shown when cells are competing, one of the cells may proteolyze their own proteins in unusual ways to kill the invading cells; such is the case of human cells when these are invaded by bacteria [33]. The presence of embedded AMPs within proteins has been recognized to be found also in the yeast, *Saccharomyces cerevisiae*, which uses these AMPs to eliminate competing yeast during wine fermentation [34]. These results have motivated the identification of AMPs generated by proteolysis in yeast [30]. Our method may become a useful tool to assist such research projects aimed to identify novel AMPs embedded in proteins. Why are AMPs embedded in proteins? Are these AMP sequences selected, or are they just by chance? These and many other questions may require further studies that would help us understand the basis for this phenomenon. Our tool may be used to explore some of these questions, as we have exemplified analyzing the relationship between AMP frequency and longevity in rodents. 

The relevance of identifying novel AMPs embedded in proteins is not limited to the role of killing microbial cells. Some AMPs may induce the activation of autophagy, which is a fundamental cellular process that affects aging in mammals; the induction of autophagy has been shown to increase longevity in different animals [35,36]. If embedded AMPs play a role in inducing autophagy in animals, we hypothesized the longer an animal lives, the larger the arsenal of embedded AMPs this would have. To test this hypothesis, we identified the embedded AMPs found in three rodents for which the longevities and proteomes are available: naked mole rat (*Heterocephalus glaber*), common rat (*Rattus norvegicus*), and common cuy (*Cavia porcellus*). Please note, in Figure 5, all animals with known longevities are included, while in Table 1 we computed all animals for which longevities and proteomes were reported; from that table we were only able to compare rodents because these were the only ones with representatives in the low, medium, and long longevity groups. The longest-lived rodent (*H. glaber*) showed the larger number of embedded AMPs despite having less proteins and less total number of amino acids than the other rodents. The second longest-lived rodent also had a larger number of embedded AMPs than the shortest-lived rodent despite this last one having larger proteomes and number of total amino acids. This data provides support to our hypothesis. Further experiments would be required to test the relevance of enriched proteomes with embedded AMPs; for instance, proteins enriched with AMPs may be swamped between long-lived organisms and short-lived ones.

Our results do not constitute proof that any of these predicted AMPs would have an activity to kill microbes or to induce autophagy in mammalian cells. At this stage, we propose this tool may assist researchers to identify potential AMPs. Further experiments may be needed to validate these predictions. However, as noted above, the computer software described in the present work represents a novel tool to explore ideas that were not addressed by any previously available software.

## 4. Materials and Methods

### 4.1. Equipment and Libraries Used

The machine used in this work consists of a cluster of 31 nodes, including the master node using LUSTRE as the parallel file system, SLURM as the task manager, RedHat Enterprise Linux 6.7 (kernel 2.6.32-573.el6.x86_64) as the operating system, a G++ compiler version 7.2.0, Intel C++ compiler version 18.0.0, GlibC (GNU libC) version 2.12, LibSVM library version 3.22, Intel TBB library version 2018.0, 128 GB RAM memory, instructions set architecture (ISA) AVX-2, and one Intel Xeon processor E5-2670 v3 at 2.30 GHz (12 cores, 24 threads) for each node. The experiments were run on a single node of this cluster.

### 4.2. Estimating the Problem Size

The problem to solve is to identify *k*-mers within amino acids sequences; the *k*-mers range from size *i* to size *j*, and protein sequences range from size *l* to *m*. Thus, the total number of *k*-mers is computed by the formula:(2)$$\mathrm{Total}\;\mathrm{number}\;\mathrm{of}\;k\mathrm{-mers}=\sum_{p=l}^{m}\sum_{k=i}^{j}(p-k+1)$$
where *p* represents the protein length and *k* the length of *k*-mer within protein of length *p*. If the average protein size is 300 amino acid residues in length and the average number of proteins for a living organism is 10^4^, for peptides in the range of 10 to 30 residues, then for a single protein of length 300, there are 291 10-mers, 290 11-mers, until reaching 5901 *k*-mers of length from 10 to 30 for a given protein of length 300. Thus, in 10^4^ proteins we will have 5.901 × 10^7^
*k*-mers of length 10 to 30.

For each *k*-mer, it would be necessary to estimate its probability of being an AMP. Assuming 1 millisecond for each estimation, the average time to compute all possible *k*-mers for any given proteome would be 5.901 × 10^7^/(1000*60*60) = 16.639 h. Since the potential number of *k*-mers is quite large (5.901 × 10^7^), we need to also consider the time required to transform the peptides into vectors that will be used for AMP classification using ML methods. For that endeavor, it is relevant to account for the possibility that some *k*-mers will be duplicated; hence, the number of calculations would be reduced.

In the next sections we will describe how we implemented a computational solution to reduce the computation time and to deal with *k*-mers duplicity.

### 4.3. K-Mer Detection and Counting

For the identification of *k*-mers within sequences, there have been several solutions previously described in the literature [37,38,39,40,41]. All these solutions have been implemented to deal with nucleotide sequences, and all used parallel CPU but one that implemented the use of parallel GPU [40]. In all the previous works, the algorithms to handle these data included sorting and hash tables. We decided to use sorting to remove duplicates because we output a sorted list of non-duplicated *k*-mers, and sorting is faster than hashing in this double task: sorting and removing. Sorting was also less memory demanding than hashing.

Our solution to find the unique *k*-mers in a given proteome is summarized in Algorithm 1:
**Algorithm 1:***K*-mers extractor   **Input**: Proteome,            *k*_min_, //*k*-mers minimun length            *k*_max_ //*k*-mers maximum length   **Output**: unique *k*-mers   **Begin**       for i = *k*_min_ to *k*_max_ do             add(*k*-mer, proteome)       sort(*k*-mers)       Delete Duplicates   **End**

The sorting algorithm used was QuickSort [42]. This algorithm takes O(n2) in the worst case and O(nlog(n)) on average where n is the number of *k*-mers to be sorted. The pseudo code for deleting duplicated *k*-mers after sorting is described in Algorithm 2.
**Algorithm 2:** Delete Duplicates   **Input**: *k*-mers to compare   **BEGIN**       total <- length(*k*-mers)       for j = 0, k = 1 to total-1 do              if *k*-mer[j] ≠ *k*-mer[k] then                    j = j + 1                    *k*-mer[j] = *k*-mer[k]       No_of_different_*k*-mers = j   **End**

The resulting *k*-mers are stored in a linear array that contains all unique *k*-mers. The length of this resulting array is given by No_of_different_*k*-mers (see Algorithm 2). 

It is important to note that while eliminating multiple copies of the same AMP found in a proteome may reduce the computation time, this may also undermine the relevance of such peptide. Since it is not possible to anticipate the biological relevance of a peptide that is found multiple times in a protein or proteome than another found less frequently, our current implementation does not account for this trait.

### 4.4. Prediction of AMP

For each unique *k*-mer identified, we performed a prediction to decide whether the peptide belongs to the AMP class. To this aim, we used two SVM classifiers’ one was used as it is implemented in the LibSVM library [24] and the other (called in-house SVM) was built by changing the list representation into an array representation of the feature vectors so the SIMD architecture can be exploited. Both models were trained with 51 molecular descriptors computed from the peptide sequences. The SVM classifier requires the computation of 51 peptide descriptors (see Appendix A) to then follow a training process with a dataset previously described by our group [43]. The parameters used to train these SVM models were: (i) linear kernel, (ii) SVM type (-s argument): C-SVM, (iii) cost (-c argument): 0.526315789, (iv) epsilon (-e argument): 0.001, (v) lost-function epsilon (-p argument): 0.1, (vi) probability (-b argument): 1.0.

After training the SVM models, we used them to classify AMPs embedded in proteins from the shrimp transcriptome previously reported in [44]. Those peptides classified as AMPs were then used as the input for CAMPred, a code written in GO language [45] to make calls to CAMP predictor service [46]. CAMP uses four different ML algorithms: support vector machine, random forest, discriminant analysis, and artificial neural network; these architectures have already been trained with their own dataset. Only those embedded peptides classified as AMPs by both SVM and the four CAMPred methods were considered AMPs in this work. The objective, by going through a second prediction process by CAMPred, is to reduce the number of *k*-mers predicted as AMPs in the first prediction step performed by our in-house SVM model.

Notice, the accuracy of the method may be affected by the choice of the datasets used for training, as it was recently shown elsewhere [47,48]. However, the ML model used in our approach can be updated accordingly.

### 4.5. Pruning of k-Mers

To continue reducing the number of *k*-mers predicted as AMPs, we fused all contiguous *k*-mers that were predicted as AMPs into a single peptide. To do that, we implemented Algorithms 3–6. Algorithm 3 starts by sorting the *k*-mers according to their actual position in the proteome. *K*-mer[*i*].first denotes the first position of *k*-mer *i* in the same manner *k*-mer[*i*].last denotes the last position of *k*-mer *i*, in both cases with respect to the positions of the analyzed sequence in the proteome. If the *k*-mer under consideration is not contained in the current *k*-group, then it is check for adjacency or overlap (between current *k*-mer and current *k*-group); in any of the two cases, the current group final position is updated, and the next *k*-mer analyzed. If the current *k*-mer is contained within the current *k*-group, then the current *k*-group final position is not updated. Algorithm 3 is applied to each sequence in the analyzed proteome. Algorithm 4 checks whether the current *k*-mer is embedded in the current *k*-group. Algorithm 5 verifies if the current *k*-mer is adjacent to the current *k*-group. Finally, Algorithm 6 checks if a prefix of the current *k*-mer overlaps a suffix of the current *k*-group.
**Algorithm 3:** Pruning   **Input**: *k*-mers derived from a specific sequence in the proteome   **Output**: groups//groups of continuous *k*-mers within the input sequence   **BEGIN**       Sort *k*-mers by their initial position in the given proteome sequence       *total* <- LENGTH(*k*-mers)//total number of *k*-mers   create the empty set groups whose elements are k-group//groups contain *k*-group, //every fragment of the proteome covered by consecutive *k*-mers predicted as AMPs.       *j* <- 1       *k*-group.first <- *k*-mer[0].first //the initial position of the left most *k*-mer       while *j* < *total* do              *k*-group.last <- *k*-mer[*j* − 1].*last*//the last position of the *j −* 1 *k-mer*              if *k*-mer[*j*] NOT **Embedded**(*k*-group)                     if *k*-mer[*j*] **Connected**(*k*-group) OR *k*-mer[*j*] **Intersects**(*k*-group) then                            *k*-last <- *k*-mer[*j*]                     else                            Add *k*-group to groups                            *k*-group.*first* <- *k*-mer[*j*].*first*                            *k*-group.*last* <- *k*-mer[*j*].*last*   **END**

**Algorithm 4:** Embedded   **Input**: *k*-mer[*i*], *k*-mer[*j*]   **Output**: *found* //*k*-mer *i* is embedded within *k*-mer *j*   **BEGIN**       *found* <- FALSE       if *k*-mer[*i*].*first* ≥ *k*-mer[j].*first* AND *k*-mer[*i*].*last* ≤ *k*-mer[*j*].*last*              *found* <- TRUE   **END**

**Algorithm 5:** Connected   **Input**: *k*-mer[*i*], *k*-mer[*j*]   **Output**: *connected*   **BEGIN**       *connected* <- FALSE       if *k*-mer[*i*].*first < k*-mer[*j*].*first* AND *k*-mer[*i*].*last* == *k*-mer[*j*].*last* OR *k*-mer[*i*].*first* == (*k*-mer[*j*].last + 1) then              *connected* <- TRUE   **END**

**Algorithm 6:** Intersects   **Input**: *k*-mer[*i*], *k*-mer[*j*]   **Output**: *intersects*   **BEGIN**       *intersects* <- FALSE       if *k*-mer[*i*].*first <= k*-mer[*j*].*first* AND *k*-mer[*i*].*last < k*-mer[*j*].*last* then              *intersects* <- TRUE   **END**

### 4.6. Tool Availability

In silico Proteolysis (InProt) is a code written in C++ standard 14 that implements the solution described for finding unique *k*-mers from proteomes. This code uses the Intel TBB library [49] for parallelization. A version of this tool is available in the platform AMP-discover (https://biocom-ampdiscover.cicese.mx/, accessed on 4 January 2023) to allow execution at a HPC environment. The implementation deployed at the InProt platform receives as input a multi fasta file containing the proteome the user wants to explore and the limits for k value. The outputs are *q* (*lmax* − *lmin* + 1) files, one for each value of *k*; *lmax* is the length of the longest *k*-mer, while *lmin* is the length of the shortest one. Each file contains, in multi-fasta format, all *k*-mers predicted as AMPs according to the in-house SVM implementation reported in this work; if users want to use the CAMP predictors, they should use the implementation provided by the authors in their website [46]. Additionally, a file in multi-fasta format containing the pruned proteome (Algorithm 3) is also provided. The platform sets an execution task and sends it to a high-performance cluster. To have access to the results of the execution, the platform allows the user to search its task by its email. At any moment the user can see the status of its task.

### 4.7. Proteome Sequences

The seven protein sequences reported for the animals with longevity data were obtained from the Uniprot database. The protein sequences for each proteome are reported as Appendix A and at https://github.com/gdelrioifc/EmbeddedAMP (accessed on 4 January 2023).

## Figures and Tables

**Figure 1 antibiotics-12-00139-f001:**
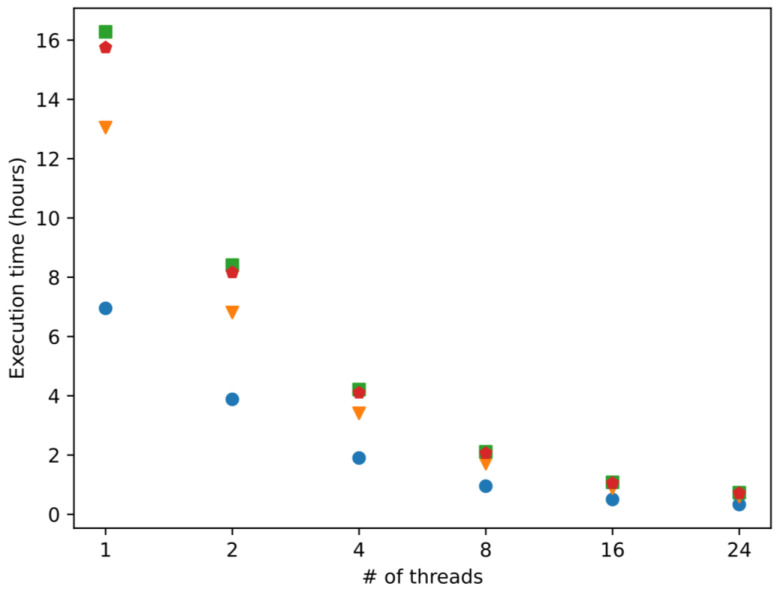
Execution times of four different implementations of the proposed algorithm. Blue circles correspond to the in-house SVM with Intel compiler. Orange triangles correspond to in-house SVM with Intel G++ compiler. Green squares represent LibSVM with Intel compiler. Red pentagons represent LibSVM with G++ compiler. # in the X axis, indicate the number of threads used in our computations.

**Figure 2 antibiotics-12-00139-f002:**
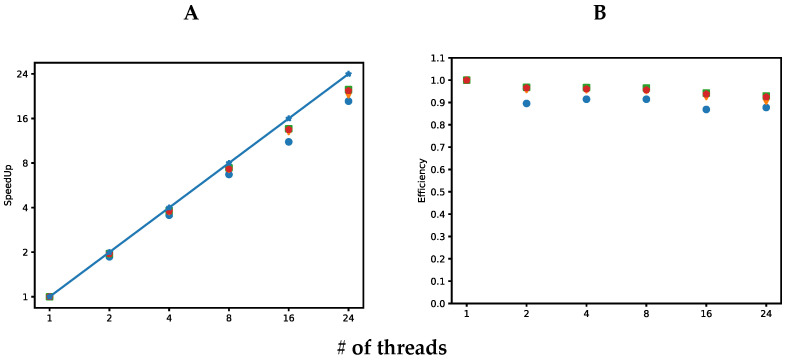
Speedups (**A**) and efficiencies (**B**) of four different implementations of the proposed algorithm. Blue circles correspond to the in-house SVM with Intel compiler. Orange triangles correspond to in-house SVM with Intel G++ compiler. Green squares represent LibSVM with Intel compiler. Red pentagons represent LibSVM with G++ compiler. The straight blue line in A represents a linear (1:1) speedup. # in the X axis, indicate the number of threads used in our computations.

**Figure 3 antibiotics-12-00139-f003:**
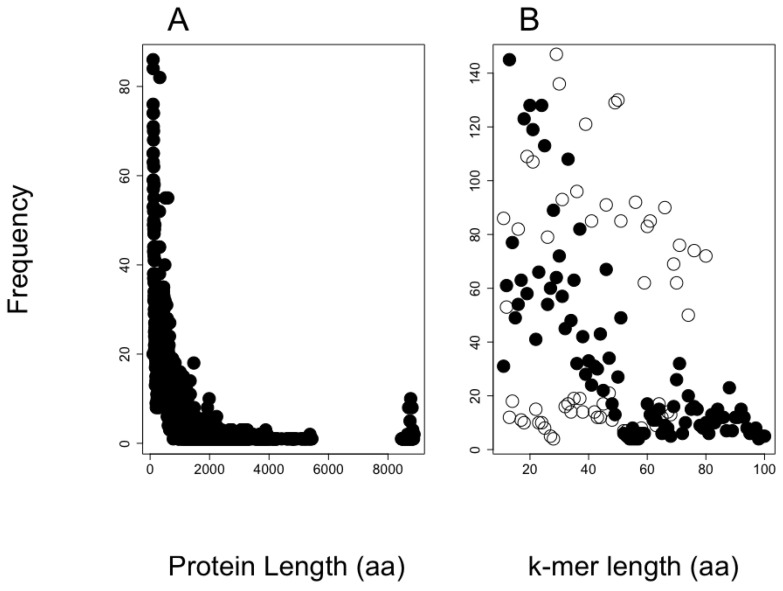
Length distribution of shrimp and AMP training set. (**A**) Protein length for the shrimp proteome and (**B**) the lengths for AMPs (black-filled circles) and non-AMPs (black empty circles) are presented with the observed frequency in the training set for our in-house SVM AMP classifier.

**Figure 4 antibiotics-12-00139-f004:**
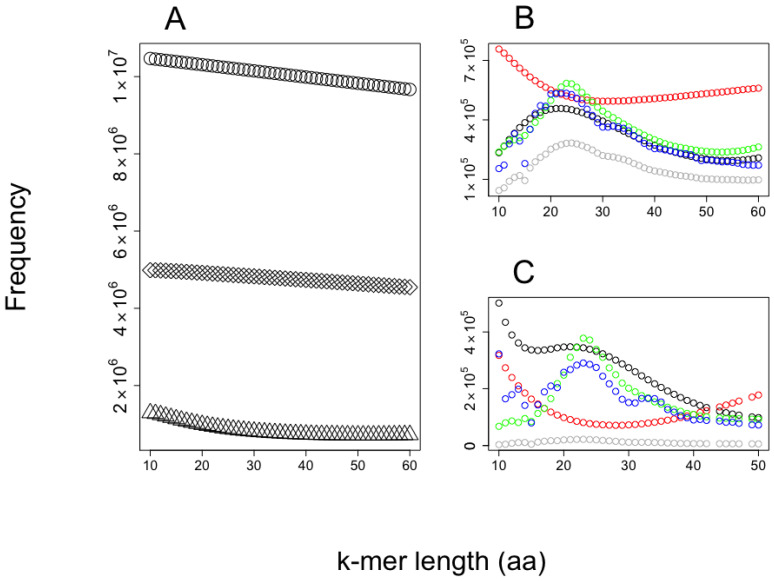
Frequencies of *k*-mers in the shrimp’s proteome. The observed frequencies of *k*-mers of different sizes were observed (**A**) while processing the shrimp’s proteome; the total number of *k*-mers are presented as circles, unique *k*-mers as rhombs, and those predicted as AMPs by our SVM implementation are indicated by triangles. For every *k*-mer predicted as AMPs in (**A**), these were analyzed by our CAMPred code to detect (**B**) antimicrobial peptides and (**C**) non-antimicrobial peptides predicted as AMPs. In this case, the four different predictors included in CAMP are reported individually: artificial neural network (red circles), random forest (blue circles), discriminant analysis (green circles), and support vector machine (black circles). (**B**,**C**) also show the intersection set for all these predictions (gray circles).

**Figure 5 antibiotics-12-00139-f005:**
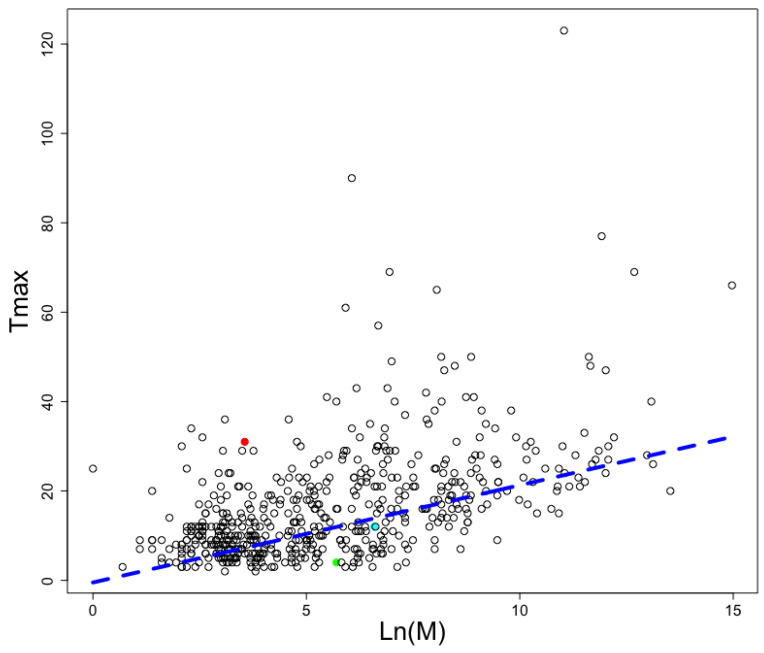
Longevities and weights in animals. The adjusted linear relationship (blue broken line) for animals reported in AnAge database. The naked mole rat (red circle), the common cuy (cyan circle), and the common rat (green circle) are shown as examples of long-, average-, and low-lived rodents. The y-axis represents the maximum number of years an animal is reported to live, and the x-axis, Log(M), represents the natural logarithmic value of the animal weight in grams.

**Table 1 antibiotics-12-00139-t001:** Animal proteomes with embedded AMPs.

Longevity	Order	Species	# Proteins	Proteome Length	# AMP	AMP/Protein Length (%)
High	Chiroptera	*Myotis lucifugus*	33,415	20,196,147	21,596	0.1069
High	Rodentia	*Heterocephalus glaber*	32,702	20,603,935	22,221	0.1078
High	Chiroptera	*Desmodus rotundus*	25,076	15,885,570	20,017	0.1260
High	Chiroptera	*Eptesicus fuscus*	37,833	26,288,993	21,689	0.0825
Medium	Rodentia	*Cavia porcellus*	28,662	18,152,925	20,838	0.1147
Low	Rodentia	*Rattus norvegicus*	34,707	23,307,820	14,078	0.0604
Low	Soricomorpha	*Condylura cristata*	23,989	15,175,763	19,986	0.1316

# represents the number of AMP or Proteins in the columns labeled # AMP or # Proteins, respectively.

## Data Availability

The protein sequences for each proteome are reported as Appendix A and at https://github.com/gdelrioifc/EmbeddedAMP, accessed on 4 January 2023.

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
