# Peer review of "Embedded-AMP: A Multi-Thread Computational Method for the Systematic Identification of Antimicrobial Peptides Embedded in Proteome Sequences"

_antibiotics, 2023, doi:10.3390/antibiotics12010139_

Round 1
Reviewer 1 Report
This manuscript reported use SVM model to predict antimicrobial peptides. This kind of approaches has been used for a long time. This paper does not contain training details of dataset, accuracy, and other ML results. There is no used to report computational speedup when one is not sure about model and predictions.
Author Response
This manuscript reported use SVM model to predict antimicrobial peptides. This kind of approaches has been used for a long time. This paper does not contain training details of dataset, accuracy, and other ML results. There is no used to report computational speedup when one is not sure about model and predictions.
Indeed, SVM and other ML-based approaches have been used to predict antimicrobial peptides (AMP). The goal of our work is not in the development of a new predictor of AMP, but instead to design an efficient and scalable in silico digestion approach to then use ML tools to identify embedded AMP within whole proteins. We hope the reviewer would agree that such innovation is contributing to the field of AMP, where so far there is no other tool available to identify embedded AMP within proteins. Furthermore, our group has developed ML-based approaches (see [15] and [16] in our list of references and [a] listed below) that are assessed with the metrics mentioned by the reviewer. We have compared Deep Learning Vs. classical ML approaches (see [16]) and self-learned Vs. handcrafted representations (see [a]). The reviewer concern is addressed in full detail in these works. To further clarify our contribution we have added the following sentence on the discussion section.
“Notice that the main contribution of our work is not the ML approach but efficient and scalable algorithms for the in silico digestion of proteomes.”
[a]. García-Jacas, C. R., García-González, L. A., Martinez-Rios, F., Tapia-Contreras, I. P., & Brizuela, C. A. (2022). Handcrafted versus non-handcrafted (self-supervised) features for the classification of antimicrobial peptides: complementary or redundant?. Briefings in Bioinformatics, 23(6), bbac428.
Reviewer 2 Report
The study provide an web server with implementation to predict antimicrobial peptides for whole proteome sequences. The program enumerated all combination with different length from 10 to 60, then sort and remove duplication. Then, the unique sequences are predicted with an in house implementation of SVM and then by CAMPred. The the number of positive predictions are reduced by removing shorter one within a longer one, or connecting the adjoined or partial overlay.
The implementation itself looks solid, while I'm very sure if the implementation are more efficient or robust than that by just trimming the prediction from long sequences, or extending prediction from short sequences with dynamic programming.
The method are applied to shrimp proteome and several rodents proteome.
Some minor points.
1. line 270, "a cluster of 31 processors", should be "a cluster of 31 nodes"
2. line 308. previous work uses "sorting and hash tables". In this work, sorting are used, is there a reason to prefer sorting then hash?
3. line 180-201 and 245-260 are not very consistence. line 251 said "3 rodents" but more are mentioned in line 180-201
4. line 354, should "lineal kernel " be "linear kernel"?
5. line 352, "51 peptide descriptors (see Table 1)", Table 1 are not peptide descriptors
Author Response
The study provide an web server with implementation to predict antimicrobial peptides for whole proteome sequences. The program enumerated all combination with different length from 10 to 60, then sort and remove duplication. Then, the unique sequences are predicted with an in house implementation of SVM and then by CAMPred. The the number of positive predictions are reduced by removing shorter one within a longer one, or connecting the adjoined or partial overlay.
The implementation itself looks solid, while I'm very sure if the implementation are more efficient or robust than that by just trimming the prediction from long sequences, or extending prediction from short sequences with dynamic programming.
Trimming the prediction from long sequences will require a predictor of AMP for long sequences that has to be built from scratch, although a very interesting idea, it is out of the scope of this work. That will require large data for such long AMP, which is currently not available. Instead, we implemented a solution where overlapping short AMP are concatenated to identify long sequences harboring AMP. Extending prediction from short sequences by using DP implies two assumptions: i) we have at hand a short seed-AMP sequence (that is what we are trying to identify), and ii) we have a reference sequence that will allow us to extend the short sequence by DP, we do not have access to this putative long AMP sequence.
The method are applied to shrimp proteome and several rodents proteome.
Some minor points.
- line 270, "a cluster of 31 processors", should be "a cluster of 31 nodes"
The reviewer is correct, it is a cluster of 31 nodes. Thank you for pointing this out. The manuscript was corrected accordingly.
- line 308. previous work uses "sorting and hash tables". In this work, sorting are used, is there a reason to prefer sorting then hash?
Yes, there is a reason to preferring sorting to hash tables, since we output a sorted list of k-mers with no duplicates, sorting does the job faster than hashing. We have added a comment to address this in page 10, lines 333 to 335 that reads as follows.
“We decided to use sorting to remove duplicates because we output a sorted list of non-duplicated k-mers and sorting is faster than hashing in this double task: sorting and removing. Sorting was also less memory demanding than hashing.”
- line 180-201 and 245-260 are not very consistence. line 251 said "3 rodents" but more are mentioned in line 180-201
We appreciate for the note. The reason for this inconsistency is since Figure 5 describes the longevity of all animals with known longevity; the motivation was to show in Figure 5 the relationship between weight and longevity. Then, in Table 1 we describe the animals with known longevity and reported proteomes; for our work we needed both data. And finally, we only used the common cuy, rat and naked mole rat, because these 3 animals all belong to the same biological order (rodentia), thus allowing for the comparison, and they all have reported longevity and proteomes. To clarify this, we have included the following phrase in the discussion section: “Please note that in Figure 5, all animals with known longevity are included, while in Table 1 we computed all animals for which longevity and proteomes were reported, from that table we were only able to compare rodents because these were the only ones with representatives in the low, medium and long longevity groups”.
- line 354, should "lineal kernel " be "linear kernel"?
Thanks for the observation. It should be “linear kernel” and the paper has been revised for consistency.
- line 352, "51 peptide descriptors (see Table 1)", Table 1 are not peptide descriptors
We appreciate the note. We forgot to include the table to describe the 51 descriptors used to train our in-house method. We are including now that table and change the reference to supplemental Table S3.
Reviewer 3 Report
Dear Authors,
Thank you very much for the opportunity to review the manuscript entitled “Embedded-AMP: A multi-thread computational method for the systematic identification of antimicrobial peptides embedded in proteome sequences”. The article describes the development of a machine learning model that uses different lengths of k-mers to identify antimicrobial peptides that may be embedded within a larger protein sequence. The idea is very interesting and novel; however, there are several points that should be taken into account before acceptance for publication. Any chance to resolve the suggestions below would be greatly appreciated. Thank you for your time and consideration.
1) Might it be possible to structure the results into subsections for easier reading? For example, “2.1 Model Description”
2) Why was only the shrimp proteome used? How did the data perform compared to the known transcriptome? The breaking down of the S. cerevisiae GAPDH was cited; please test the data on the S. cerevisiae proteome to validate the model if mentioned in the text. Otherwise, please use additional datasets to validate the predictions.
3) There are several databases that contain sequences for antimicrobial peptides in addition to CAMPred (e.g. http://dramp.cpu-bioinfor.org/ , https://dbaasp.org/ , https://aps.unmc.edu/). How would these data change the model predictions?
4) Only unique k-mers/AMPs were analyzed. Are duplications of any AMPs relevant to autophagy, AMP activity, or longevity?
5) The connection between AMPs, autophagy, longevity is quite a reach – either please provide more references as to how these are connected or select a different outcome to compare with AMP presence.
6) If longevity is still used as a metric for the effect of AMPS, please discuss why AMP/Protein length (%) is not consistent with the longevity classifications (i.e. High, Medium, Low).
7) Experimental validation of selected top-ranked predicted AMPs would be ideal, but (as mentioned in the text) this may not be possible.
8) The website looks great and easy to use, although the data limitation (only up to 1MB) is quite restricting (I realize this may be difficult to increase from a funding/data accessibility standpoint).
Author Response
Dear Authors,
Thank you very much for the opportunity to review the manuscript entitled “Embedded-AMP: A multi-thread computational method for the systematic identification of antimicrobial peptides embedded in proteome sequences”. The article describes the development of a machine learning model that uses different lengths of k-mers to identify antimicrobial peptides that may be embedded within a larger protein sequence. The idea is very interesting and novel; however, there are several points that should be taken into account before acceptance for publication. Any chance to resolve the suggestions below would be greatly appreciated. Thank you for your time and consideration.
1) Might it be possible to structure the results into subsections for easier reading? For example, “2.1 Model Description”
In this new version we have added subsections to facilitate the reading of our work.
2) Why was only the shrimp proteome used? How did the data perform compared to the known transcriptome? The breaking down of the S. cerevisiae GAPDH was cited; please test the data on the S. cerevisiae proteome to validate the model if mentioned in the text. Otherwise, please use additional datasets to validate the predictions.
We appreciate the comment. In this new version, we included the predictions on the previously reported AMP identified from the proteolysis of S. cerevisiae as well as on its GAPDH. Our results show that our method can correctly predict 20 out of the 33 AMP embedded in yeast proteins. We have included a new results section to describe these findings and included 2 supplemental tables.
3) There are several databases that contain sequences for antimicrobial peptides in addition to CAMPred (e.g. http://dramp.cpu-bioinfor.org/ , https://dbaasp.org/ , https://aps.unmc.edu/). How would these data change the model predictions?
Yes, there are many databases for AMP, like those mentioned by the reviewer and some newer ones. Our group has gathered most of the databases currently available in one (see [11]). The results in the prediction will change with a different database, we proved this in our recent work [b] and so does the work of Sidorczuk et al. [c]. Thanks to the reviewer observation we add a comment in the paper addressing this issue (see page 12, lines 400 to 402). Notice however, that this situation does not affect the contribution of our work that is the efficient and scalable in silico digestion to identify AMPs within proteins.
“Notice that the accuracy of the method may be affected by the choice of the datasets used for training, as it was recently shown elsewhere [49,50]. However, the ML model used in our approach can be updated accordingly.”
[b] Pinacho-Castellanos, S. A., García-Jacas, C. R., Gilson, M. K., & Brizuela, C. A. (2021). Alignment-free antimicrobial peptide predictors: improving performance by a thorough analysis of the largest available data set. Journal of Chemical Information and Modeling, 61(6), 3141-3157.
[c] Sidorczuk, K., Gagat, P., Pietluch, F., Kała, J., Rafacz, D., Bąkała, L., ... & Burdukiewicz, M. (2022). Benchmarks in antimicrobial peptide prediction are biased due to the selection of negative data. Briefings in Bioinformatics, 23(5), bbac343.
4) Only unique k-mers/AMPs were analyzed. Are duplications of any AMPs relevant to autophagy, AMP activity, or longevity?
This is a valid question, and we appreciate the note. Indeed, we may anticipate that having multiple times an AMP embedded within a protein may increase the amount of the duplicated AMP, which in turn may increase its concentration and activity as well. However, not every duplicated AMP will be present at the same time (these duplicated peptides may be present in different proteins that may or may not present at the same time or the same cellular location or may be generated by different protease cuts). To clarify this, we have added the following sentence in section 4.3 : “It is important to note that, while eliminating multiple copies of the same AMP found in a proteome may reduce the computation time, this may also undermine the relevance of such peptide. Since it is not possible to anticipate the biological relevance of a peptide that is found multiple times in a protein or proteome than another found less frequently, our current implementation does not account for this trait”.
5) The connection between AMPs, autophagy, longevity is quite a reach – either please provide more references as to how these are connected or select a different outcome to compare with AMP presence.
Excellent point. We included in the results section, a recent work reviewing AMP that induce autophagy and a recent review regarding the autophagy-longevity relationship: “Considering that AMP have been associated with autophagy induction [39] and that autophagy induction has been shown to increment lifespan in many different species [40], we decided to analyze the proteomes of mammals with reported longevity…”.
6) If longevity is still used as a metric for the effect of AMPS, please discuss why AMP/Protein length (%) is not consistent with the longevity classifications (i.e. High, Medium, Low).
We appreciate the note. We have noted that the frequency of embedded AMP in different rodents is not related with the proteome length, but instead is a property of the protein composition. In consequence, it is not expected that the AMP/protein length ratio will be related to the longevity classifications. Instead, the AMP frequency is the trait that relates to longevity. To clarify this, we have included in the results section 2.5 the following sentence: “In consequence, the AMP/Protein length ratio is not related to longevity, only the AMP frequency”.
7) Experimental validation of selected top-ranked predicted AMPs would be ideal, but (as mentioned in the text) this may not be possible.
We appreciate the comment. In this new version, we included the predictions on the previously reported AMP identified from the proteolysis of S. cerevisiae as well as on its GAPDH. Our results show that our method can correctly predict 20 out of the 33 AMP embedded in yeast proteins. We have included a new results section to describe these findings and included 2 supplemental tables.
8) The website looks great and easy to use, although the data limitation (only up to 1MB) is quite restricting (I realize this may be difficult to increase from a funding/data accessibility standpoint).
Thank you for pointing out this relevant point, we have updated our website so it can support larger files, as required to analyze larger proteomes.
Reviewer 4 Report
The authors used an in-silco digested proteome to identify peptides with plausible antimicrobial activity. This is an interesting study. However, experimental validation is critical to such studies. While there are certain limitations to validate a large array of peptides identified, I suggest the authors to compare the potential peptide hits against, previously reported AMPs from the same organisms / or other AMP data bases as a way to validate their method. Ideally the authors are likely to get several peptides matching the reported AMPs.
Author Response
The authors used an in-silco digested proteome to identify peptides with plausible antimicrobial activity. This is an interesting study. However, experimental validation is critical to such studies. While there are certain limitations to validate a large array of peptides identified, I suggest the authors to compare the potential peptide hits against, previously reported AMPs from the same organisms / or other AMP data bases as a way to validate their method. Ideally the authors are likely to get several peptides matching the reported AMPs.
We appreciate the comment. In this new version, we included the predictions on the previously reported AMP identified from the proteolysis of S. cerevisiae as well as on its GAPDH. Our results show that our method can correctly predict 20 out of the 33 AMP embedded in yeast proteins. We have included a new results section to describe these findings and included 2 supplemental tables.
Round 2
Reviewer 3 Report
Dear Authors,
Thank you for the opportunity to review the updates to your manuscript. Thank you very much for including the points from the previous review into the new version. Overall, the manuscript looks good! A couple of minor points for potential improvement are listed below. Any consideration in taking these into account would be greatly appreciated. Congratulations on the novel and interesting work. Thank you for your time.
1) Website still says 1MB
2) On the y-axes of Figures 1 and 2, what are the units?
3) Can the false-positive rate for the two tests in Section 2.6 also be reported?
Author Response
1) Website still says 1MB
We appreciate the note. We want to note though, that we have updated our website. Perhaps the reviewer must reload the webpage to note the change we have done (?). Please let us know if the problem persists. We noted that the 1MB message appeared in cell phone browsers, we are currently working to address this issue, however, standard browsers in a PC should not have any problem.
2) On the y-axes of Figures 1 and 2, what are the units?
We appreciate the note. Speedup and efficiency do not have any units, which are the 2 performance measures reported in Figure 2. Please note that Speedup is now defined as “S_t=T_1/T_t, where T_1 is the execution time of the implementation for 1 thread, while T_t is the execution time of the implementation for t threads”, while “Efficiency measures the amount proportion of time that a processor is effectively used and is defined as E_t=S_t/t, where S_t is the speedup achieved for t threads”.
In reviewing the other figures, we realized that the legends for figures 1 and 3 were wrong and have corrected them in the revised version of our work.
3) Can the false-positive rate for the two tests in Section 2.6 also be reported?
This is an important aspect in evaluating the reliability of a predictor. We are now including the false-positive rates observed in the two test sets included in section 2.6. We modified the text to report the true-positive rate as well.